# Vari-Focal Light Field Camera for Extended Depth of Field

**DOI:** 10.3390/mi12121453

**Published:** 2021-11-26

**Authors:** Hyun Myung Kim, Min Seok Kim, Sehui Chang, Jiseong Jeong, Hae-Gon Jeon, Young Min Song

**Affiliations:** 1Gwangju Institute of Science and Technology, School of Electrical Engineering and Computer Science, 123 Cheomdangwagi-ro, Buk-gu, Gwangju 61005, Korea; gusaud31@gist.ac.kr (H.M.K.); seok9643@gmail.com (M.S.K.); shchangj@gm.gist.ac.kr (S.C.); 2SOSLAB, B-101, BI Center, GIST 123 Cheomdangwagi-ro, Buk-gu, Gwangju 61005, Korea; stopstar@soslab.co; 3Gwangju Institute of Science and Technology, AI Graduate School, 123 Cheomdangwagi-ro, Buk-gu, Gwangju 61005, Korea

**Keywords:** light field camera, micro-lens array, vari-focal lens, depth estimation

## Abstract

The light field camera provides a robust way to capture both spatial and angular information within a single shot. One of its important applications is in 3D depth sensing, which can extract depth information from the acquired scene. However, conventional light field cameras suffer from shallow depth of field (DoF). Here, a vari-focal light field camera (VF-LFC) with an extended DoF is newly proposed for mid-range 3D depth sensing applications. As a main lens of the system, a vari-focal lens with four different focal lengths is adopted to extend the DoF up to ~15 m. The focal length of the micro-lens array (MLA) is optimized by considering the DoF both in the image plane and in the object plane for each focal length. By dividing measurement regions with each focal length, depth estimation with high reliability is available within the entire DoF. The proposed VF-LFC is evaluated by the disparity data extracted from images with different distances. Moreover, the depth measurement in an outdoor environment demonstrates that our VF-LFC could be applied in various fields such as delivery robots, autonomous vehicles, and remote sensing drones.

## 1. Introduction

Camera systems have advanced to mimic the mechanism and function of biological eyes [1,2,3,4]. However, conventional 2D imaging systems are limited in capturing the real world because of the lack of depth information. A light field camera provides a facile way to capture 3D information (light field or plenoptic function) of an object with only a single image sensor and micro-lens array without any external light source, so it has attracted considerable interest in both academia and industry. A micro-lens array (MLA), a core optical component, acting like a multi-camera array, acquires light field data which can be utilized in depth estimation, multi-viewing imaging, digital refocusing, and 3D reconstruction. The initial light field cameras were proposed with two different approaches: placing a pinhole grating inside the camera; placing a micro-lens in front of the image plane [5,6]. Based on these concepts, Ng introduced the first hand-held plenoptic camera called the standard light field camera (plenoptic 1.0) in 2005 [7], and a focused light field camera (plenoptic 2.0) was proposed by Lumsdaine and Georgiev in 2009 [8]. Since then, numerous studies have flourished based on these two types of light field camera and various applications have been explored by utilizing the light field data. For instance, the development of MLA fabrication in a large area facilitates the integration of customized light field cameras for all-in focus imaging from a single exposure [9], and geometrical calibration and depth estimation algorithms have been developed to extract depth information from the lenslet light field camera [10,11].

In particular, 3D depth sensing is one of the major concerns in modern security or automation systems, and the light field camera could be a vital technique for depth sensing vision systems [12]. To realize 3D imaging equivalent to human vision, near- (~1 m) to mid-range (~10 m) depth sensing is required in the artificial vision system including the light field camera. However, a conventional light field camera is inherently limited in shallow depth of field (DoF) because the MLA has a finite DoF. To address this problem, extensive research has been conducted to extend the DoF, including the adoption of multi-focal MLA [13,14,15] and focal length modulation based on liquid-crystal MLA (LC-MLA) [16,17]. Although multi-focal MLA and LC-MLA successfully extend the DoF to some extent, there still exist difficulties in practical application because of their complex manufacturing process, low spatial resolution in multi-focal MLA, inconsistent focal length tendency at low voltage and/or relatively low transparency in LC-MLA [18,19,20,21]. Therefore, another approach to extend the DoF in the main lens rather than MLA should be explored.

In this study, we demonstrate the vari-focal light field camera (VF-LFC) with extended DoF for near to mid-range depth sensing applications. Each optical component, such as a vari-focal lens, and MLA is selected and designed to meet the target DoF based on numerical calculations presented in the following section. We considered both the DoF of MLA (image plane) and the DoF of the main lens (object plane), to optimize the focal length of the MLA. For mechanical robustness of the system, an optical alignment system was designed by ray-tracing simulation considering the space between the image sensor and cover glass, and both cover glass thickness and MLA thickness. In addition, an alignment system was designed to be adaptable with the c-mount optical components, which is generally adopted in machine vision. The implemented VF-LFC successfully extracts depth information within the range of 0.5 m to 15 m by varying its focal length according to the region of interest. To verify the extended DoF of the fabricated VF-LFC, the simulated and measured disparity data according to object distance were analyzed. In addition, we demonstrated the measurement in an outdoor environment to confirm that our VF-LFC was applicable to various fields. The proposed VF-LFC verifies that the DoF customization of the LFC system can be readily achieved by optimizing the focal lengths of MLA and vari-focal lenses.

## 2. Materials and Methods

Conventional light field cameras consist of a fixed focal lens as a main lens, the MLA, and image sensor as shown in Figure 1a. Each micro-lens of the MLA acts as a single camera by focusing the image from the main lens on the image sensor with slightly different perspectives, so that it is possible to capture light field data including disparity. The disparity should be guaranteed to have a sufficient slope to extract depth information within the distance range, but it rapidly converges as the distance increases out of the DoF of the light field camera, as shown in Figure 1c. Therefore, to extract depth information from far objects, a light field camera should have a large DoF. However, in the conventional light field camera system with fixed focal lens, the MLA has a fundamentally limited DoF, so the conventional light field camera suffers from a shallow DoF [22].

To overcome this limitation, we adopted the vari-focal lens as a main lens which can adjust its focal length within the range of 20 mm to 75 mm (CBC co., Ltd., Computar lenses, Tokyo 309084, Japan) instead of the fixed focal lens. By changing the focal length of the system, the total DoF of the system can be largely extended, while providing sufficient disparity slope, as shown in Figure 1d. Here, we divided four measurement regions with four specific focal lengths (20 mm, 30 mm, 50 mm, and 75 mm) to achieve depth estimation with an object distance of up to 15 m. In the measurement region of short focal length, the depth information of near objects can be extracted, and as the focal length increases, it is available to estimate the depth of the faraway objects. It is noted that each measurement region should be partially overlapped with successive measurement regions to realize continuous depth estimation over a total DoF of the system. In addition, each measurement region should have a large disparity change for the depth estimation with high accuracy.

The VF-LFC consists of a vari-focal lens as main lens, a micro-lens array, and an image sensor as shown in Figure 2a. Based on plenoptic camera 2.0, an image of an object located at $a_{L}$ is formed by the main lens at the position of $b_{L}$. The MLA is located behind the image plane with a distance of a, and the image sensor is located behind the MLA with a distance of b. The MLA focuses the image of the main lens, as an object, on the image sensor. The depth estimation principle of the VF-LFC is as follows. The distance $a_{L}$ from the main lens to the object can be calculated by the thin lens equation using the focal length of the main lens $f_{L}$ and $b_{L}$, which is expressed by
(1)$$\frac{1}{f_{L}}=\frac{1}{a_{L}}+\frac{1}{b_{L}}$$

Each micro-image formed on the image sensor shows the main lens image with a slightly different point of view. Figure 2b describes the triangulation scheme based on the projection of two micro-images. $p_{x1,2}$ is the distance between the image point to the principal point of the respective micro-image, and $d_{1,2}$ is the distance from the intersection point of the image plane and a baseline that is an optical axis of the respective micro-lens. In Figure 2b, an up-pointing arrow and a down-pointing arrow of distance present positive and negative values, respectively. Triangles which have equal angles are similar, and thus, the following relation is validated:(2)$$p_{xi}=\frac{d_{i}\cdot b}{a}$$

The baseline distance between two micro-lenses is defined as follows:(3)$$d=d_{2}-d_{1}$$

Similarly, if we define the parallax of the image point, $p_{x}$, as the difference between $p_{x1}$ and $p_{x2}$, it can be defined as follows from Equations (2) and (3):(4)$$p_{x}=p_{x1}-p_{x2}=\frac{(d_{2}-d_{1})\cdot b}{a}$$

The distance between the image plane and the MLA, a, is described as a function of the baseline distance d, the distance between the MLA and sensor, b, and the estimated parallax, $p_{x}$
(5)$$a=\frac{d\cdot b}{p_{x}}$$

In addition, the distance between the image plane and main lens, $b_{L}$, can be calculated by a and known parameters $B_{L}$ and b.
(6)$$b_{L}=B_{L}-b-a$$

The object plane distance from the main lens, $a_{L}$, is calculated by substituting Equation (5) into the Equation (1), which is expressed by
(7)$$a_{L}={\left(\frac{1}{f_{L}}-\frac{1}{B_{L}-b-a}\right)}^{-1}$$

Consequently, the depth can be estimated from a point which is simultaneously focused on at least two micro-images.

Figure 2c shows the image side DoF according to the MLA focal length. $F_{L}$ is a focal length of the main lens when the object is at infinity. If the image side DoF exceeds the $F_{L}$, the image become blurred because it is out of the focal range. Therefore, the MLA focal lengths of 955 µm and 965 µm are more desirable than 975 µm for the selected vari-facal lens. For the depth estimation with high accuracy, the object should be located within the DoF of the light field camera system. Since the image side and object side DoFs are correlated with each other, the object side DoF is determined by the image side DoF, $a_{+}$ and $a_{-}$. The object side DoF was calculated for focal lengths of 20 mm, 30 mm, 50 mm, and 75 mm, respectively (Figure 2d). The dotted lines indicate the boundaries between measurement regions of each focal length. In case of the focal length 955 µm, the object side DoF does not cover the target object distance of 15 m, even though there is no blurred region within the image side DoF. In this regard, we designed the MLA of VF-LFC to have a focal length of 965 μm considering the DoF of both the image side and the object side.

## 3. Results and Discussion

Figure 3a shows the design schematic of an optical alignment module according to the target MLA focal length, and the integrated VF-LFC module is shown in Figure 3b. We set the C-mount flange back distance of 17.5 mm and the image plane distance with the consideration of the main lens focal point of 5.3 mm. Given the previously designed MLA focal length of 965 µm and the MLA center of DoF distance 4.5 mm, the distance b from the MLA to the image sensor was set to 1.235 mm through the thin lens equation in Equation (1). Figure 3c shows the measurements of near and far objects with focal lengths of 20 mm and 75 mm through the proposed VF-LFC. For a near object, a focused light field image was acquired by the focal length 20 mm, whereas in the case of focal length 75 mm, the micro-image itself, which lowers accuracy during image processing, is too large and blur occurs. On the other hand, for a far object, the micro-image of the object is too small to distinguish and shows low resolution when the focal length is 20 mm. Therefore, the focal length 75 mm is more adequate for the far object imaging, and its micro-image is shown as similar to the micro-image of the near object captured with the focal length 20 mm. As a result, Figure 3c clearly shows that the object DoF can be extended by controlling the focal length according to the target distance. The parameters of the MLA and image sensor are presented in Table 1.

In Figure 4, the experimental demonstration was conducted by VF-LFC to verify that the disparity variation within each measurement region for different focal length was sufficient to extract depth information with high accuracy. The checkerboard calibration target (with downsized checkerboard in the center) was used as an object for light field imaging from near to far distance, as shown in Figure 4a. The object distance for each focal length was divided as follows: FL20 is 0.5–1.5 m at 0.5 m intervals; FL30 is 1–5 m, FL50 is 4–10 m; and FL75 is 8–15 m at 1 m intervals. For continuity between adjacent measurement regions, the measurement region for each focal length was divided to contain partially overlapping portions with neighboring measurement regions. Figure 4c shows a portion of the reference image captured at each focal length. Although the captured images are measured with different focal lengths, all captured images have the region that shows similar disparity with those from other measurement regions, which indicates that the VF-LFC was finely aligned and assembled as initially designed. Figure 4d shows the disparity graph along the object distance based on the calibration result and experimental result. The solid line shown in the graph is based on the calculation of Equation (6), and parallax was converted to disparity, which is a pixel unit. The intrinsic parameters used in the calculation are presented in Table 2. From the image captured by VF-LFC, the disparity was calibrated by fixing the internal variable to suit our device based on the open source light field tool box 2.0 [23]. Disparity calculation was performed on adjacent micro-lens images through a semi-global matching process, and disparity values were extracted for each of 10 locations from the boundary line of the checkerboard for acquiring accurate values. The average and standard deviations of the calculated disparity values are expressed as red dots and error bars, respectively. Even though the standard deviation of measured data seems relatively large as the disparity slope is steeper, the overall trend of measured data is well matched with the calibration results. As a result, the sufficient disparity variation is guaranteed up to a distance of 15 m, which indicates that our VF-LFC with greatly extended DoF can extract the depth information with high accuracy. Figure 4e shows the sequential depth measurement scene with signs at five different distances. Micro-images from four different focal lengths (Figure 4f) are stitched after image rendering, and then, the stitched image is overlapped by the normalized disparity map as shown in Figure 4g. It clearly indicates that the proposed VF-LFC can extract sequential depth information within the extended DoF range. Table 3 classifies the light field camera systems into objective lens, measurement range and features. It shows that our VF-LFC achieves the most extended DoF with a simple configuration compared to other light field cameras regardless of the light field type.

To demonstrate the practicality of the VF-LFC, the outdoor measurement was performed, as shown in Figure 5. Measurements were carried out at four different focal lengths. Regardless of focal length, clear micro-images were acquired even outdoors, as shown in Figure 5b. Figure 5c describes the relationship among the field of view, object distance and focal length of a vari-focal lens. As the focal length increases, the field of view is naturally narrowed down to clearly focus the farther objects. Figure 5d shows the rendered images by rendering algorithms (top row) and disparity maps based on extracted data from micro-images with a VF-LFC at each focal length. It should be noted that each disparity map at different focal lengths shows continuous disparity variation, which indicates that high accuracy depth information can be estimated within the entire extended DoF, up to 15 m of object distance. Hence, the proposed vari-focal light field camera enables the acquiring of depth information from the near to far area compared to conventional light field camera systems. Further advancement in disparity map extraction can be achieved by additional post-processing to reduce the disparity map artefacts caused by excessive light reflection in the scene [28,29].

## 4. Conclusions

In summary, we proposed the VF-LFC with an extended DoF by introducing a vari-focal lens as a main lens. Unlike conventional light field cameras, which suffer from an inherently limited DoF, the proposed VF-LFC successfully extends its DoF up to ~15 m. We divided the four measurement regions according to the focal lengths of a vari-focal lens (20 mm, 30 mm, 50 mm, and 75 mm) to achieve the target DoF. Each measurement region contains the partially overlapping area with its adjacent region for continuity. By considering both the DoF (image side) of MLA and the DoF (object side) of the main lens, the focal length of MLA was designed to be 965 μm. For mechanical robustness, a light field module was precisely designed to meet an optical alignment, and the fabricated module was adaptable to a C-mount lens, which is generally used for machine vision. The disparity data extracted from the object images according to each measurement region are well matched with the theoretically calculated values and indicate that the fabricated VF-LFC can provide depth information with the range of the extended DoF. In addition, the outdoor measurement demonstrates that our VF-LFC can be a practical technique for various applications such as autonomous vehicles, delivery robots, and remote sensing drones that require 3D information acquisition from short to medium distances.

## Figures and Tables

**Figure 1 micromachines-12-01453-f001:**
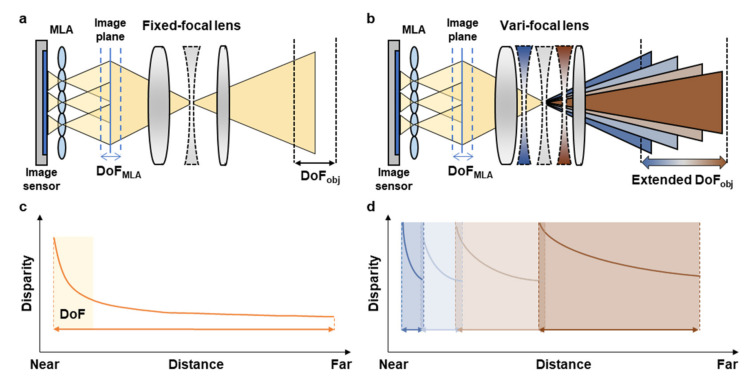
(**a**,**b**) Schematic of conventional light field camera (**a**) and proposed light field camera with a vari-focal lens (**b**). (**c**,**d**) Relationship between disparity and object distance. (**c**) The disparity of the conventional light field camera decreases and converges. (**d**) By controlling the focal length before the disparity converges, the proposed VF-LFC provides an extremely extended DoF.

**Figure 2 micromachines-12-01453-f002:**
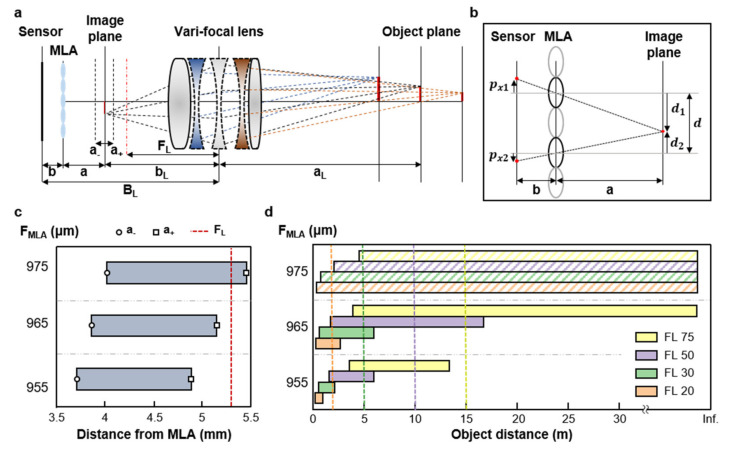
(**a**) Schematic of the VF-LFC system with main design parameters. (**b**) Schematic of the optical path through the MLA between the image sensor and the image plane of the main lens. (**c**) The image side DoF, $a_{+}$ and $a_{-}$, are calculated from the thin lens equation for a focal length of MLA. (**d**) The object side DoF for each focal length of a vari-focal lens according to the MLA focal length, the dotted lines are the boundaries between measurement regions of each focal length.

**Figure 3 micromachines-12-01453-f003:**
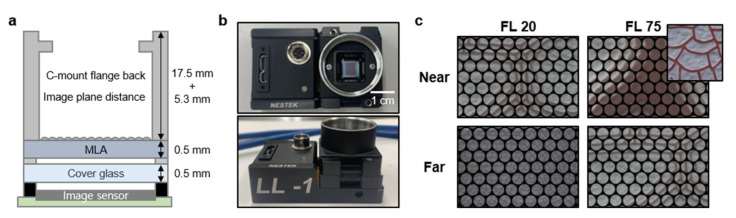
(**a**) Schematic of optical alignment module for VF-LFC. (**b**) Photographs of the assembled module for VF-LFC (**c**) Photographs of light field image captured by VF-LFC with focal length FL 20 mm (left), and FL 75 mm (right), inset: original texture image.

**Figure 4 micromachines-12-01453-f004:**
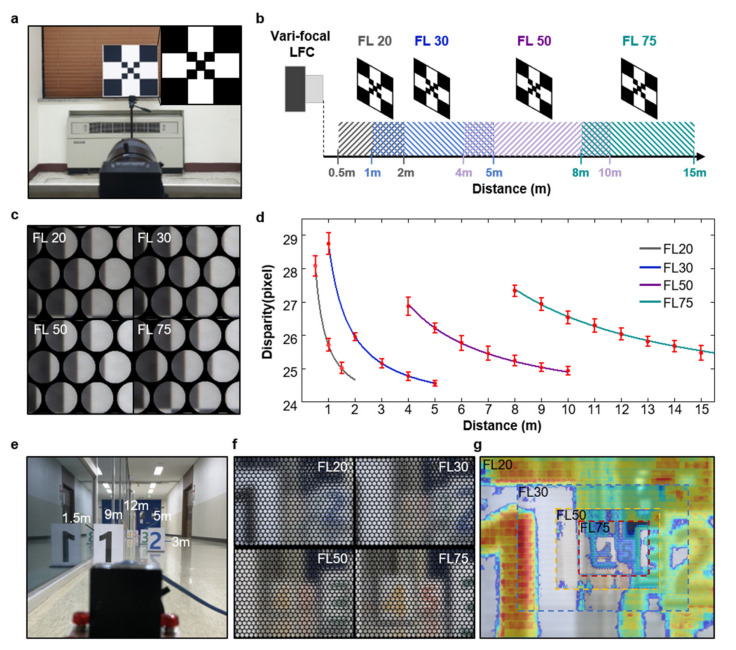
(**a**) Photograph of the VF-LFC and reference image (inset). (**b**) Schematic of VF-LFC experimental setup according to the measurement regions for each focal length. (**c**) Micro-images from VF-LFC at four different focal lengths. (**d**) Graph of the disparity according to object distance, solid lines are calibrated result and red dots represent experimental result. (**e**) Photograph of sequential depth measurement setup with the VF-LFC. (**f**) Micro-images from VF-LFC at four different focal lengths. (**g**) Stitched image with four different focal length photographs, overlapped by normalized disparity map.

**Figure 5 micromachines-12-01453-f005:**
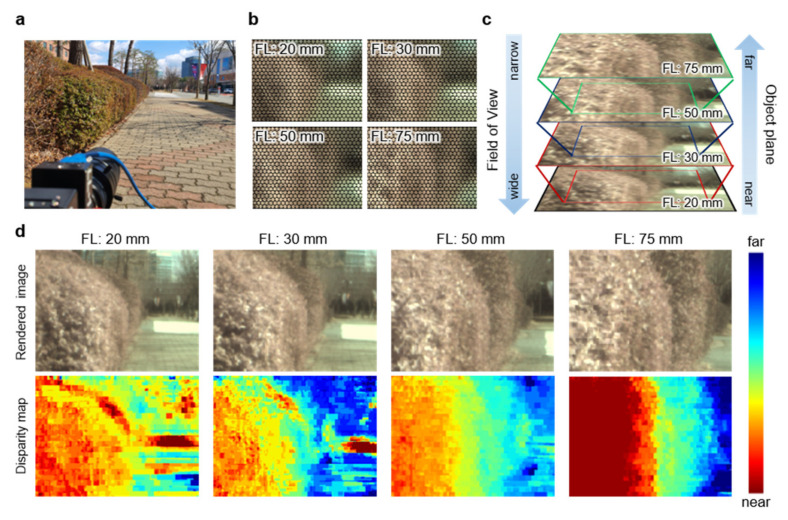
(**a**) Photograph of outdoor measurement setup of VF-LFC. (**b**) Microimages from VF-LFC at four different focal lengths. (**c**) Photographs indicating the field of view difference and object distance for each focal length. (**d**) Photographs after image rendering (top row) and disparity maps according to each focal length captured by a VF-LFC (bottom row).

**Table 1 micromachines-12-01453-t001:** Parameters of MLA and image sensor.

Microlens Array		Image Sensor	
**Pitch**	240 µm	Model	IMX178 (Sony, Tokyo, Japan)
**Focal length**	965 µm	Pixel size	2.4 µm
**Array type**	Hexagonal	Number of pixels	3096 (H) × 2080 (V)

**Table 2 micromachines-12-01453-t002:** Intrinsic parameters for vari-focal light field camera.

b (mm)	d (μm)	$f_{L}\;(mm)$	$b_{L}\;(mm)$
1.235	101.8	20, 30, 50, 75	5.3$-b+f_{L}$

**Table 3 micromachines-12-01453-t003:** Light field camera classification.

Light Field Type	Objective Lens	Depth of Field	Main Characteristics	Reference
Unfocused LFC (1.0)	Zoom lens	0.05–2 m	Lytro 1st generation	[24]
Unfocused LFC (1.0)	Zoom lens	0.5–0.9 m	-	[25]
Focused LFC (2.0)	3.04 mm	0.05–0.25 m	Small form factor	[26]
Focused LFC (2.0)	35 mm	0.7–5.2 m	Raytrix camera	[27]
Focused LFC (2.0)	39.8 mm	~0.59 m	LC MLA (tunable)	[19]
Focused LFC (2.0)	Zoom lens	0.5–15 m	Vari-focal lens	Ours

## Data Availability

Not applicable.

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
