# Peer review of "Vari-Focal Light Field Camera for Extended Depth of Field"

_micromachines, 2021, doi:10.3390/mi12121453_

Round 1

Reviewer 1 Report

The manuscript of micromachines-1457329 reported a vari-focal light field camera (VF-LFC) with the extended DoF for mid-range 3D depth sensing applications. As a main  lens of the system, a vari-focal lens with four different focal lengths is adopted to extend the DoF up to about 15 m. This research is technically sound, but I suggest that the manuscript should be modified and enhanced in content as following.

1. The detailed information of vari-focal lens should be given, including the manufacturer, model and core parameters.
2. The detailed parameters of MLA and inmage sensor are missing.
3. The authors should check serial number of the references.
4. The innovation of the paper needs to be further clarified in the section of introduction.

Author Response

We sincerely appreciate the reviewer for valuable and critical comments that were helpful to improve the quality of our manuscript significantly. We modified our manuscript according to the reviewer’s comments.

Reviewer 2 Report

In this work, to achieve LF camera with higher DoF, the authors proposed to add 2 more focal lens to have vari-focal lens (20mm, 30mm, 50mm, and 75mm); Then, it sets up measurement parameters to achieve higher DoF for LF images. The technical contribution is sound. However, there are some major concerns: 

1) Why a set of focal lens (20mm, 30mm, 50mm, and 75mm) is adopted ? In fact, to satisfy the human observation (to see difference of DoF) , a set of focal lens in common camera is 20, 50, 75, 105 ?

2) DoF can be affected by explicit factors (not focal) such as: the distance from camera to object, how do the authors solve this ?

3) When changing the depth of DoF image, we need compare and evaluate the depth, how do the authors select the algorithms to compute the depth ? there is currently several ways to measure the depth with different results. 

4) The current presentation of paper is not clear enough both in structure and explanation. The authors should add more (even brief) descriptions on device analysis, setup scenarios, background and related works
\scenarios, Background and related work.

Author Response

(The authors gave the same response as above.)

Round 2

Reviewer 1 Report

All the comments and suggestions are addressed, and I recommend the manuscript to be accepted in present form.

Reviewer 2 Report

The authors have addressed all of my comments!